# Controlling Nutritional Status (CONUT) Score as a Predictive Marker in Hospitalized Frail Elderly Patients

**DOI:** 10.3390/jpm13071119

**Published:** 2023-07-10

**Authors:** Aurelio Lo Buglio, Francesco Bellanti, Cristiano Capurso, Gianluigi Vendemiale

**Affiliations:** Department of Medical and Surgical Sciences, University of Foggia, 71122 Foggia, Italy; aurelio.lobuglio@unifg.it (A.L.B.); francesco.bellanti@unifg.it (F.B.); cristiano.capurso@unifg.it (C.C.)

**Keywords:** malnutrition, frailty, hospitalization, hospitalized elderly, CONUT score

## Abstract

The Controlling Nutritional Status (CONUT) score is a simple screening tool able to detect altered nutritional status as well as to predict clinical adverse outcomes in specific populations. No data are available in frail patients. This study aims to investigate the predictive role of the CONUT score on mortality and length of stay (LOS) in frail patients admitted to an Internal Medicine Department. We consecutively enrolled 246 patients aged 65 years or older, divided into two groups based on frailty status. The two groups were further divided according to low (<5) or high (≥5) CONUT score. Length of stay (LOS) was higher in frail patients than not-frail patients, as well as in the frail group with high CONUT scores compared to the frail group with low CONUT scores. Multiple linear regression showed an increase of 2.1 days for each additional point to the CONUT score. In-hospital mortality was higher in frail compared to not-frail patients, but it did not differ between frail patients with high CONUT scores and frail patients with low CONUT scores. An analysis of the survival curve for 30-day mortality showed a higher mortality rate for frail/high-CONUT-score patients as compared to the not-frail/low-CONUT-score group. The CONUT score shows high prognostic value for higher LOS—but not mortality—in the clinical setting of internal medicine departments for old frail patients.

## 1. Introduction

Malnutrition shows high prevalence among hospitalized patients, ranging from 20% to 50%, with peaks of 90% in hospitalized elderly patients [1]. Malnutrition is directly linked with increased morbidity and mortality in older people and represents a major cause of late recovery after long in-hospital stay, especially in the presence of underlying diseases [2,3]. Since malnutrition is associated with prolonged stay in hospital, controlling nutritional status plays a key role in determining several clinical outcomes [4]. Malnutrition is also associated with frailty, an important geriatric syndrome, which is highly prevalent among hospitalized elderly patients and associated with several hospital-related unfavorable clinical outcomes such as disability, excessive length-of-stay (LOS), and mortality [5,6,7]. 

Nutritional status can be assessed using several screening tools [8]. Controlling Nutritional Status (CONUT) is an easy, user-friendly, and reproducible nutritional screening score based on serum albumin, total cholesterol concentration and total lymphocyte count, which has been shown to predict mortality in hospitalized elderly patients [9,10,11]. It was first presented as an immune-nutritional score—a screening tool allowing the automatic daily assessment of nutritional status of all inpatients that undergo routine analysis [12]. The CONUT score has a high prognostic value in specific populations such as the elderly, cancer patients, or patients affected by heart failure or ischemic stroke [10,13,14,15].

Nevertheless, the predictive role of the CONUT score in hospitalized frail patients has not been established yet. This study aimed at assessing CONUT at admission as a predictive score of hospital outcomes such as LOS, in-hospital stay, and 30-day mortality in frail elderly patients.

## 2. Materials and Methods

### 2.1. Study Population and Design

The study was conducted at the Department of “Medicina Interna e dell’Invecchiamento”, “Policlinico Riuniti” in Foggia (Italy), as a prospective and observational single-center study, based on a population aged 65 years or older consecutively recruited from January 2022 to October 2022. The exclusion criteria were the following: dysphagia, active cancer, severe cognitive impairment (assessed with a Mini-Mental State Examination [MMSE] score ≤ 9 points), prolonged in-hospital stay caused by nursing home placement, and inability to comply with the study protocol or to provide written informed consent. While the MMSE was routinely performed on all hospitalized patients, dysphagia was assessed in selected patients by a trained logopedics professional.

Patients were divided into two groups (not frail and frail) based on the presence of frailty, defined according to Fried’s criteria as described previously [16]. Patients that met at least three of five criteria were defined as frail, whereas patients presenting with less than three criteria were defined as not frail. Patients were further divided into two subgroups—high CONUT and low CONUT, according to the presence of a CONUT score ≥ 5 or <5, respectively.

### 2.2. Biochemical Analysis and Data Collection

A blood sample was taken at admission, to determine haemoglobin, white blood cells (WBCs), lymphocytes, glucose, albumin, total cholesterol, creatinine, triglycerides, and C reactive protein (CRP). Height and body weight were measured according to standardized procedures. Body weight in bedridden patients was measured using a LinkoScale 350 system (LikoAB, Luleå, Sweden). Body mass index (BMI) was calculated as the ratio between body weight and square height in meters. 

### 2.3. Multidimensional Assessment

Nutritional status was evaluated through the Mini Nutritional Assessment (MNA) and CONUT score. MNA is an 18-item questionnaire concerning anthropometric, general, dietetic, and subjective evaluation. MNA identifies patients with malnutrition (score < 17), at risk of malnutrition (score 17–23.5), and that are well-fed (score > 23.5) [17].

The CONUT score is calculated based on serum albumin (g/dL), total lymphocyte count (count/mm^3^), and total cholesterol (mg/dL), as reported in Table 1 [12]. Cognitive status was assessed using the Mini Mental State Examination (MMSE), a validated tool that assesses orientation, memory, attention, ability to follow verbal instructions and produce written language, and visuospatial skills [18]. 

Function autonomy was measured using the activities of daily living (ADL) scale and the instrumental activities of daily living (IADL) scale. ADL evaluates basic daily activities, such as bathing, getting dressed or eating; the IADL scale considers complex activities, such as use of public transportation, finances, or shopping. ADL score ranges from 0 to 6 points, while the IADL score ranges from 0 to 8 points [19]. Depressive symptoms were evaluated using the Geriatric Depression Scale short-form (GDS-SF), a 15-item tool validated in elderly people, which classifies subjects into the following categories: No depression (0 to 5 points), mild depression (6 to 9 points), and severe depression (10 to 15 points) [19,20].

### 2.4. Length of Stay and Mortality

Length of stay (LOS), defined as the time between the patient’s admission to the hospital and discharge, was recorded. Additionally, the number of deaths were recorded during hospitalization and at 30 days of hospital admission (index admission) by medical record or by telephone call in the case of discharge before 30 days.

### 2.5. Statistical Analysis

Continuous variables were expressed as mean ± standard deviation of the mean (SD) or median (Interquartile Range, IR) and analyzed using Student’s *t*-test or Mann–Whitney’s test. Nominal and categorical variables were expressed as *n* (%) and analyzed using the Chi-Square test or Fisher’s exact test. Parametric or non-parametric distribution was evaluated by the Kolmogorov–Smirnov test. 

The impact of frailty as well as the presence of a CONUT score ≥ 5 on LOS was analyzed by two-way analysis of variance (ANOVA). Post-hoc analysis was performed using the Tukey test. 

Correlation analysis was performed using Pearson or Spearman coefficient according to the variable distribution. 

The multiple linear regression analysis using a center model was performed to study the effect of independent variables (age, gender (F), MNA score, MMSE score, ADL, IADL, GDS, C-reactive protein (CRP) and CONUT score) on LOS as the dependent variable. Survival analysis was performed using Kaplan–Meier curves, and comparisons were made using the log-rank statistic. Furthermore, to estimate the effect of independent variables on time to discharge, a Cox proportional hazard model was used. A sample size of 120 in each group was required to detect the difference at 90% power and 5% level of significance. Statistical tests were performed using Statistical Package for Social Sciences version 20.0 (SPSS, Inc., Chicago, IL, USA) software analysis, and a two-sided *p* value < 0.05 was considered significant. Graphs were built using the GraphPad Prism 8.0 for Windows (GraphPad Software Inc., San Diego, CA, USA).

## 3. Results

We enrolled 246 patients, including 147 females (59.8%), with an average age of 77.7 ± 8.0 years. Patients were divided in two groups based on frailty status: 121 (49.2%) were included in the not-frail group and 125 (50.8%) were in the frail group. Table 2 reports admitting diagnosis, while baseline characteristics are summarized in Table 3.

There were no significant differences among frail and not-frail patients as regards genre, number of comorbidities, serum levels of glucose and triglycerides. 

Frail patients were older with lower values of haemoglobin, lymphocytes, albumin, and total cholesterol. Conversely, they presented significantly higher WBCs, creatinine, and CRP as compared to not-frail patients (Table 2). The frail group showed lower BMI than the not-frail group, as well as worse nutritional status, cognitive performances, and functional autonomy. A higher CONUT score and the prevalence of depressive symptoms were recorded in the frail group with respect to the not-frail group. Furthermore, higher mortality and LOS were reported in the frail group with respect to the not-frail group (Table 3). The deaths recorded in the hospital were related to heart failure (4, 21%), respiratory failure (4, 21%), and acute infectious diseases (11, 58%).

The prevalence of co-morbidities at hospital admission is summarized in Table 4. The most common comorbidities in both groups were hypertension, diabetes, chronic obstructive pulmonary disease (COPD) and chronic kidney disease (CKD), defined as an estimated Glomerular Filtration Rate (e-GFR) less than 60 mL/min/1.73 m^2^ at the time of admission. No differences were found among low- and high-CONUT groups both in not-frail and in frail patients.

Table 5 summarizes data of frail patients stratified according to the CONUT score. Patients presenting with a high CONUT score were mostly females and had lower concentrations of blood lymphocytes, serum albumin and total cholesterol. Additionally, CRP median values were higher in the high-CONUT than the low-CONUT group. No significant differences were found between these groups regarding age, comorbidities, haemoglobin, WBCs, glucose, creatinine, and triglycerides.

Interestingly, no differences were found among the two groups about nutritional status evaluated with MNA, cognitive and functional status or prevalence of depression symptoms. 

In-hospital and 30-day mortality rates were found not to be different between frail patients with low and high CONUT scores as well as between not-frail patients with low and high CONUT scores (Table 6, Figure 1). However, frail patients with high CONUT scores showed a significantly higher LOS (Table 6). A two-way ANOVA, with LOS as the dependent variable and frailty and CONUT score as independent factors, was performed, showing that the presence of frailty (F = 5.605, *p* 0.019) and a COlNUT score ≥ 5 (F = 70.735, *p* < 0.001) influenced the LOS, with no interaction between these two variables (F = 2.236, *p* 0.136). Post-hoc analysis demonstrated that frail and not-frail patients with high CONUT scores had significantly higher LOS compared to frail or not-frail patients with low CONUT scores. No differences between frail and not-frail patients with low CONUT scores or between frail and not-frail groups with high CONUT scores were found (Figure 2).

In the frail group, we performed correlation analysis corrected by gender and age that showed a significant negative correlation between LOS and MMSE score or ADL (*p* 0.025 and 0.012, respectively), and a positive correlation with CONUT score (*p* < 0.001). No significant correlation was found among LOS and CRP, IADL, MNA, and GDS score (Table 7). 

The significant variables in the correlation analysis were used to build a multiple linear regression model, using LOS as the dependent variable. Of these variables, the CONUT score was significantly associated with LOS, with a coefficient of 2.11 (*p* < 0.001, IC95% 1.78–2.40) (Table 8).

A comparison among survival curves for 30-day mortality showed a significant difference between frail–high CONUT and not-frail–low CONUT (mean survival days 26.9 ± 7.3 vs. 29.6 ± 2.5, *p* < 0.001). The mean survival for not-frail–high CONUT and frail–low CONUT were 29.0 ± 4.5 and 28.7 ± 4.8 days, respectively (Figure 3).

We then applied a Cox proportional hazard (PH) model to estimate the effect of independent variables on time to discharge. The univariate analysis identified high MNA, ADL, and IADL and low CONUT scores as being associated with longer LOS, but only IADL and CONUT scores significantly influenced LOS in the multivariate model (Table 9). As shown in Figure 4, LOS was significantly higher in frail patients with high CONUT scores rather than in the other groups studied.

## 4. Discussion

The major finding of this study consists in the association between high CONUT scores on admission and length of stay in hospitalized frail elderly patients. The prevalence of frailty among our study population (50.8%) was higher than in previous studies [5,21]. Frail patients showed significantly worse nutritional status as well as lower cognitive and functional performance. Additionally, depressive symptoms were prevalent among frail patients with respect to not-frail patients. Frailty is a geriatric syndrome that affects multiple domains, especially among hospitalized patients [5,22]. In this scenario, frailty, per se, is associated with poor clinical outcomes [23]. The results demonstrated that frailty predicts prolonged length of stay both in medical and surgical patients [24,25,26]. Our study confirms this observation since LOS was higher in frail rather than not frail patients. Other than frailty, malnutrition plays a crucial role on clinical outcomes in hospitalized patients [27]. The CONUT score has been described as a very useful tool in identifying hospitalized patients at risk of malnutrition in different populations, such as elderly patients [28]. Ulibarri et al. found a significant association between CONUT and the Subjective Global Assessment (SGA) or the Full Nutritional Assessment (FNA) in hospitalized patients, regardless of age. Rinninella et al. described a significant correlation between CONUT classes and nutritional status, evaluated with both the Nutritional Risk Screening 2002 (NRS-2002) and the Malnutrition Universal Screening Tool (MUST) in a population aged 18 years and older [29]. Nevertheless, Cabré et al. reported a poor agreement between CONUT score and MNA in hospitalized elderly patients [30]. In our study, more than 70% of frail patients presented with a high CONUT score, but the two groups with low (<5) or high (≥5) CONUT scores had a similar MNA score, suggesting that this tool could fail in detecting malnutrition in frailty. This finding suggests that specific validation studies are needed to determine the reliability of CONUT in old frail patients.

The results demonstrated that the CONUT score predicts poor clinical outcomes in different acute settings, such as oncological, cardiac, and surgical pathologies [11,14,31,32]. In a systematic review and meta-analysis, Takagi et al. showed that a high CONUT score was a prognostic factor associated with poor overall survival and a higher rate of recurrence in colorectal cancer patients [33]. Recently, Liu et al. found an association between a high CONUT score and the risk of higher LOS when compared to a low CONUT score in elderly hospitalized patients. Furthermore, a higher CONUT score was predictive of in-hospital mortality [10]. Interestingly, this study reports that LOS was higher in frail patients with high CONUT scores as compared to those with low CONUT scores. Frailty is characterized by reduced resilience and more difficult recovery, with consequent prolonged LOS, especially in acute hospital settings [34,35]. Similarly, malnutrition is closely associated to an increased risk of prolonged LOS and increases the risk of developing frailty through long-term insufficient protein and energy intake [36,37,38]. Interestingly, we found an independent effect of both frailty and high CONUT score on LOS. In particular, we found a close correlation between CONUT score and LOS, with 2.1 more days of hospitalization for each additional point of the CONUT score. Furthermore, the frail group showed a higher rate of in-hospital mortality (*p* 0.002) compared to not-frail patients. Even though the CONUT score demonstrated the highest predictive ability among different nutritional tools for in-hospital mortality in elderly hospitalized patients [10], we could not report any significant difference in mortality between frail patients stratified according to the CONUT score.

The analysis of survival curves showed the increased mortality of frail patients with high CONUT scores compared to those who are not frail and with low CONUT scores. However, further studies involving a larger number of patients are needed. 

To the best of our knowledge, this is the first study to investigate the association of CONUT score with clinical characteristics in a population of frail hospitalized elderly patients. However, this study has several limitations. First, this is a single-center study with a longitudinal design, so there is no statement of causality. Second, the small sample size restricted the subgroup analysis, and the heterogenic distribution of the sample size could have affected statistical analysis. Third, we did not record the number of patients living in a nursing home prior to hospitalization. Moreover, residual confounding factors such as the presence of coexisting diseases (such as COPD and diabetes) and their severity were not evaluated in the interaction model between CONUT and LOS. Finally, this study may have presented some bias since it was performed in a single center, so the results may not be generalized to other populations. Nevertheless, although further multicenter studies are needed to verify our results, this investigation represents an important step in understanding the association between CONUT score and clinical outcomes in hospitalized patients. 

## 5. Conclusions

In conclusion, the CONUT score shows a high prognostic value for higher LOS in the clinical setting of internal medicine departments for old frail patients. These data suggest the possible use of the CONUT score as a nutritional screening tool for identifying frail patients with a higher risk of protracted hospitalizations. The predictive role of different CONUT score cut-off values needs to be validated in populations with different diseases in future multi-center, large-sample, prospective clinical studies.

## Figures and Tables

**Figure 1 jpm-13-01119-f001:**
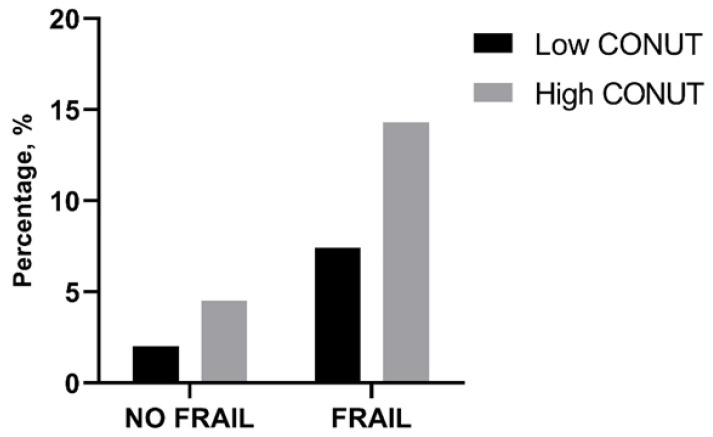
Percentage of in-hospital mortality in frail and not-frail group according to low- or high-CONUT group. CONUT, controlling nutritional status. Statistical differences were assessed by Fisher’s exact test.

**Figure 2 jpm-13-01119-f002:**
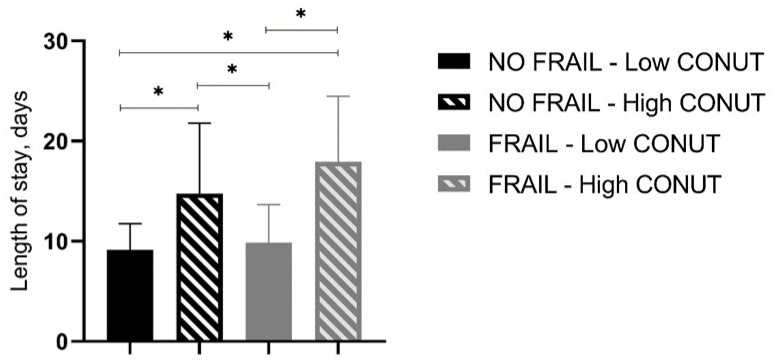
Length of stay (days) in all groups of studied patients. CONUT, controlling nutritional status. Statistical differences were assessed by two-way ANOVA and Tukey’s as post-hoc test. * *p* < 0.001.

**Figure 3 jpm-13-01119-f003:**
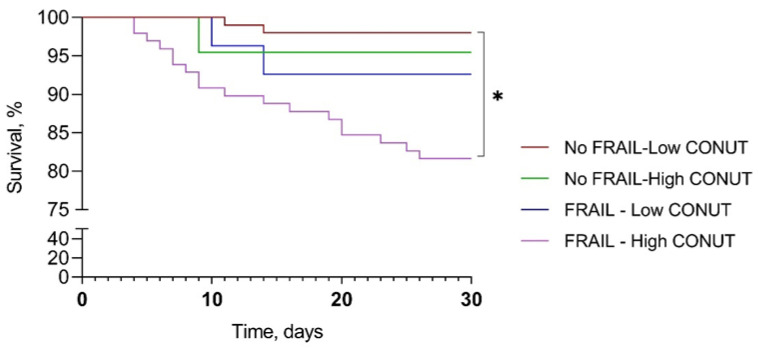
Kaplan–Meier survival curve according to frailty and CONUT score. CONUT, controlling nutritional status. * *p* < 0.001.

**Figure 4 jpm-13-01119-f004:**
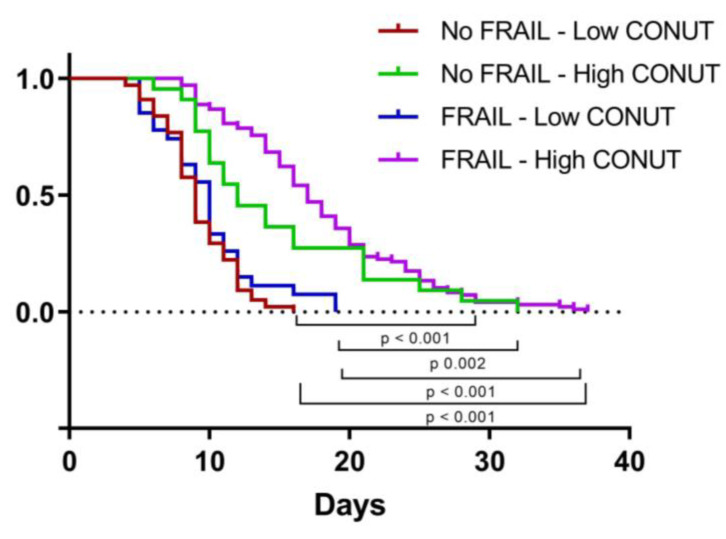
Kaplan–Meier curve representing time to discharge according to frailty and CONUT score. CONUT, controlling nutritional status.

**Table 1 jpm-13-01119-t001:** CONUT score calculation.

	Nutritional Status			
Variables	Normal	Light	Moderate	Severe
Albumin (g/dL) Score	≥3.5 0	3.0–3.49 2	2.5–2.9 4	<2.5 6
Total lymphocyte (/mm^3^) Score	>1600 0	1200–1599 1	800–1199 2	<800 3
Total cholesterol (mg/dL) Score	>180	140–180	100–139	<00
Screening total score	0–1	2–4	5–8	9–12

Abbreviation: CONUT, controlling nutritional status.

**Table 2 jpm-13-01119-t002:** List of admitting diagnosis.

Respiratory failure	67 (27.2%)
Sepsis	44 (17.9%)
Heart failure	30 (12.2%)
Cirrhosis	13 (5.4%)
Renal failure	10 (4.1%)
Pleural effusion	10 (4.1%)
Pneumoniae	8 (3.2%)
Other conditions	64 (25.9%)

**Table 3 jpm-13-01119-t003:** Characteristics of patients and nutritional, cognitive, and functional performance according to frailty status.

	Not Frail N = 121 (49.2%)	Frail N = 125 (50.8%)	*p* Value
Age, years	74.6 ± 6.4	80.7 ± 8.3	**<0.001**
Genre F	67 (55.4%)	80 (64.0%)	0.168
Co-morbidities > 3	48 (39.7%)	62 (49.6%)	0.117
Haemoglobin, g/dL	12.4 ± 2.1	11.2 ± 1.8	**<0.001**
WBCs, *n*/mm^3^	6700 [5285, 8550]	8070 [5245, 11,500]	**0.026**
Lymphocytes, *n*/mm^3^	1617 [1271, 2254]	989 [705, 1413]	**<0.001**
Glucose, mg/dL	104 [88.0, 128.5]	110.5 [83.7, 145.2]	0.458
Albumin, g/dL	3.5 ± 0.5	2.8 ± 0.5	**<0.001**
Total cholesterol, mg/dL	169.9 ± 44.9	138.1 ± 43.7	**<0.001**
Creatinine, mg/dL	0.9 [0.7, 1.1]	1.2 [0.8, 104]	**0.006**
Triglycerides, mg/dL	111.0 [78.5, 152.5]	104.0 [78.0, 143.5]	0.279
CRP, ng/ml	5.2 [1.8, 16.6]	29.4 [9.9, 47.4]	**<0.001**
BMI, Kg/m^2^	28.2 ± 5.5	26.6 ± 5.3	**0.028**
MNA, score	23.5 ± 4.2	18.6 ± 4.9	**<0.001**
CONUT, score	3 [1, 4]	6 [5, 8]	**<0.001**
MMSE, score	22.6 ± 6.9	17.7 ± 9.0	**<0.001**
ADL, score	6 [5, 6]	4 [1, 6]	**<0.001**
IADL, score	5 [4, 8]	2 [1, 5]	**<0.001**
GDS-SF, score	3 [1, 6]	5 [3, 7]	**0.001**
LOS, days	10.2 ± 4.3	16.2 ± 6.9	**<0.001**
In-hospital Mortality, *n* (%)	3 (2.5%)	16 (12.8%)	**0.002**

Data are expressed as mean (± standard deviation), median [interquartile range] or *n* (percentage) as appropriate. Abbreviations: M, male; F, female; WBCs, white blood cells; CRP, C reactive protein; MNA, mini nutritional assessment; CONUT, controlling nutritional status; MMSE, mini mental state examination; ADL, activity of daily living; IADL, instrumental activity of daily living; GDS-SF, geriatric depression scale short form; LOS, length of stay. *p* value < 0.05 were considered statistically significant (in bold).

**Table 4 jpm-13-01119-t004:** Co-morbidities on hospital admission in frail and not-frail patients according to low- or high-CONUT group.

	NOT FRAIL 121 (49.2%)			FRAIL 125 (50.8%)		
	Low CONUT *n*. 99 (81.8%)	High CONUT *n*. 22 (18.2%)	*p* Value	Low CONUT *n*. 27 (21.6%)	High CONUT *n*. 98 (78.4%)	*p* Value
**Hypertension**	68 (68.7%)	12 (54.5%)	0.205	18 (66.7%)	52 (53.1%)	0.207
**Diabetes**	29 (29.3%)	7 (31.8%)	0.815	10 (37.0%)	36 (36.7%)	0.977
**Heart failure**	27 (27.3%)	5 (22.7%)	0.662	8 (29.6%)	30 (30.6%)	0.922
**Ictus**	12 (12.1%)	4 (18.2%)	0.448	9 (33.3%)	33 (33.7%)	0.974
**IHCD**	26 (26.3%)	6 (27.3%)	0.923	9 (33.3%)	30 (30.6%)	0.787
**Atrial fibrillation**	17 (17.2%)	6 (27.3%)	0.275	7 (25.9%)	29 (29.6%)	0.710
**COPD**	43 (43.4%)	9 (40.9%)	0.829	10 (37.0%)	40 (40.8%)	0.723
**CKD**	61 (61.6%)	12 (54.5)	0.540	16 (59.3%)	50 (51.0%)	0.448
**Cirrhosis**	7 (7.1%)	3 (13.6%)	0.312	3 (11.1%)	13 (13.3%)	0.767

Data are expressed as *n* (percentage). IHCD, ischemic heart chronic disease; COPD, chronic obstructive. pulmonary disease; CKD, chronic kidney disease.

**Table 5 jpm-13-01119-t005:** Characteristics of patients and nutritional, cognitive, and functional performance according to frailty status.

	Low CONUT N = 27 (21.6%)	High CONUT N = 98 (78.4%)	*p* Value
Age, years	80.1 ± 8.7	80.8 ± 8.3	0.703
Genre F	4 (14.8%)	41 (41.8%)	**0.010**
Co-morbidities > 3	12 (44.4%)	50 (51.0%)	0.545
Haemoglobin, g/dL	11.6 ± 1.8	11.1 ± 1.8	0.163
WBCs, *n*/mm^3^	8600 [6020, 12,380]	7445 [5075, 11,350]	0.339
Lymphocytes, *n*/mm^3^	1370 [860, 1910]	968 [661, 1318]	**0.012**
Glucose, mg/dL	113 [86, 143]	110 [82, 147]	0.993
Albumin, g/dL	3.3 ± 0.4	2.7 ± 0.4	**<0.001**
Total cholesterol, mg/dL	176.8 ± 50.7	127.3 ± 35.0	**<0.001**
Creatinine, mg/dL	1.1 [0.7, 1.8]	1.2 [0.7, 1.5]	0.938
Triglycerides, mg/dL	102.5 [76.7, 123.2]	106.0 [78.0, 144.0]	0.661
CRP, ng/ml	16.4 [4.6, 42.7]	32.3 [12.8, 51.7]	**0.022**
BMI, Kg/m^2^	28.8 ± 6.6	25.9 ± 4.7	**0.014**
MNA, score	19.8 ± 4.3	18.2 ± 5.1	0.184
MMSE, score	19.2 ± 8.5	17.4 ± 9.2	0.424
ADL, score	4 [1, 6]	4 [1, 6]	0.372
IADL, score	2 [1, 4]	2 [0, 5]	0.796
Barthel, score	60 [38, 82]	55 [25, 90]	0.997
GDS-SF, score	4 [2, 8]	5 [3, 7]	0.716

Data are expressed as mean (±standard deviation), median [interquartile range] or *n* (percentage) as appropriate. Abbreviation: M, male; F, female; WBCs, white blood cells; CRP, C reactive protein; MNA, mini nutritional assessment; CONUT, controlling nutritional status; MMSE, mini mental state examination; ADL, activity of daily living; IADL, instrumental activity of daily living; GDS-SF, geriatric depression scale short form. *p* value < 0.05 were considered statistically significant (in bold).

**Table 6 jpm-13-01119-t006:** Length of stay and mortality in frail and not-frail patients according to low- or high-CONUT group.

	NOT FRAIL 121 (49.2%)			FRAIL 125 (50.8%)		
	Low CONUT *n*. 99 (81.8%)	High CONUT *n*. 22 (18.2%)	*p* Value	Low CONUT *n*. 27 (21.6%)	High CONUT *n*. 98 (78.4%)	*p* Value
**LOS, Days**	9.2 ± 2.6	14.8 ± 7.0	<0.001	9.8 ± 3.8	17.9 ± 6.5	**<0.001**
**In-Hospital** **Mortality, *n* (%)**	2 (2%)	1 (4.5%)	0.455	2 (7.4%)	14 (14.3%)	0.519
**30-Day Mortality, *n* (%)**	2 (2%)	1 (4.5%)	0.455	2 (7.4%)	18 (18.4%)	0.239

Data are expressed as mean (±standard deviation), median [interquartile range] or *n* (percentage) as appropriate. LOS, length of stay; CONUT, controlling nutritional status. *p* value < 0.05 were considered statistically significant (in bold).

**Table 7 jpm-13-01119-t007:** Correlation analysis between LOS and CRP, MMSE, MNA, ADL, IADL, GDS and CONUT score corrected by gender and age.

	*R*	*p* Value
CRP, ng/mL	0.02	0.868
MNA, score	−0.04	0.731
MMSE, score	−0.23	**0.025**
ADL, score	−0.25	**0.012**
IADL, score	−0.11	0.298
GDS-SF, score	0.16	0.190
CONUT, score	0.461	**<0.001**

Abbreviations: MNA, mini nutritional assessment; MMSE, mini mental state examination; ADL, activity of daily living; IADL, instrumental activity of daily living; GDS-SF, geriatric depression scale short form; CONUT, controlling nutritional status. *p* value <0.05 were considered statistically significant (in bold).

**Table 8 jpm-13-01119-t008:** Single and multiple linear regression of factors associated to length of stay in frail patients.

Single Linear Regression Analysis				Multiple Linear Regression Analysis		
	Coefficients	*p* Value	IC 95% of Coefficient	Coefficients	*p* Value	IC 95% of Coefficient
MMSE, score	0.70	<0.001	0.50–0.81	0.11	0.175	−0.01–0.27
ADL, score	3.07	<0.001	2.40–3.64	0.09	0.794	−0.78–0.60
CONUT, score	2.38	<0.001	2.22–2.55	2.11	**<0.001**	1.78–2.40

Data are expressed as mean (±standard deviation), median [interquartile range] or *n* (percentage) as appropriate. Abbreviations: MMSE, mini mental state examination; ADL, activity of daily living; CONUT, controlling nutritional status. *p* value < 0.05 were considered statistically significant (in bold).

**Table 9 jpm-13-01119-t009:** Factors associated with length of stay by univariate Cox PH model (left columns) and multivariable Cox PH model (right columns).

Univariate Cox PH Model				Multivariable Cox PH Model		
	B Coeff.	HR (95% IC)	*p* Value	B Coeff.	HR (95% IC)	*p* Value
CRP, ng/mL	−0.009	0.971 (0.979, 1.002)	0.111			
MNA, score	0.073	1.076 (1.018, 1.137)	**0.010**	−0.44	0.957 (0.870, 1.052)	0.363
MMSE, score	0.017	1.017 (0.984, 1.051)	0.323			
ADL, score	0.110	1.116 (1.002, 1.243)	**0.046**	0.001	1.001 (0.844, 1.188)	0.991
IADL, score	0.190	1.209 (1.091, 1.340)	**0.001**	0.196	1.217 (1.042, 1.421)	**0.013**
GDS-SF, score	−0.056	0.945 (0.890, 1.004)	0.066			
CONUT, score	−0.229	0.795 (0.716, 0.883)	**0.001**	−0.223	0.800 (0.711, 0.900)	**<0.001**

Abbreviations: CRP, C-reactive protein; MNA, mini nutritional assessment; MMSE, mini-mental state examination; ADL, activities of daily living; IADL, instrumental activities of daily living; GDS-SF, geriatric depression scale short form; CONUT, controlling nutritional status.

## Data Availability

The study data are available on reasonable request to the corresponding authors.

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
