# Peer review of "Controlling Nutritional Status (CONUT) Score as a Predictive Marker in Hospitalized Frail Elderly Patients"

_jpm, 2023, doi:10.3390/jpm13071119_

Round 1
Reviewer 1 Report
This is a single center case-control study. The researches compared two groups of patients depending on their frailty status - regarding the Fried Phenotype.
The main result is increased mortality in individuals with frailty and malnutrition.
Comment:
The authors should mention why the patients were hospitalised and what other co-morbidities were present on admission.
For instance, a high rate of hypertension, diabetes, COPD, and heart failure can be expected that have an impact on mortality as well. The authors mention a comorbidity index behind which numerous different diseases can be hidden. Please clarify this.
Did the author conduct a sample size calculation and a power calculation as well? Since patient dies in hospital, would the authors also specify the main reason of death, for instance heart failure, pneumonia etc.. These data should be available.
Furthermore, what do we know about the living conditions of the patients? How many patients were living in a nursing home or were admitted newly to a nursing home. This is an important issue, since waiting for nursing home placement can explain longer LOS. Please adjust for this parameter in your analysis.
n.a.
Author Response
Comment:
The authors should mention why the patients were hospitalised and what other co-morbidities were present on admission.
For instance, a high rate of hypertension, diabetes, COPD, and heart failure can be expected that have an impact on mortality as well. The authors mention a comorbidity index behind which numerous different diseases can be hidden. Please clarify this.
Reply: According to the reviewer’s request, we mentioned admission causes (lines 129-133 and table 2) and co-morbidities in the revised version of the manuscript (lines 154-164 and table 4).
Comment:
Did the author conduct a sample size calculation and a power calculation as well? Since patient dies in hospital, would the authors also specify the main reason of death, for instance heart failure, pneumonia etc.. These data should be available.
Reply: Sample size calculation at 90% power is now reported in the manuscript (lines 120-122), and data on main causes of death were also described (lines 151-153).
Comment:
Furthermore, what do we know about the living conditions of the patients? How many patients were living in a nursing home or were admitted newly to a nursing home. This is an important issue, since waiting for nursing home placement can explain longer LOS. Please adjust for this parameter in your analysis.
Reply: We agree with this interesting observation of the reviewer. Nevertheless, data on living conditions prior of hospitalization were not recorded. We added this point as a limitation of our study (lines 316-317). Importantly, we did not consider patients who prolonged their stay pending discharge to a nursing home. This has been added in the manuscript as an exclusion criterion (line 54).
Reviewer 2 Report
The authors conducted an observational study to examine the association of Controlling Nutritional Status (CONUT) score with length-of-stay and in-hospital mortality in frail older patients. By recruiting a total of 246 patients from a hospital, the authors showed that a higher CONUT score at admission was associated with a higher length of stay in frail patients. There are some comments.
Comments:
1. Materials and Methods (1. Study Population): " - as a prospective and observational single center study, based on a population aged 65 years or older recruited from –."The exclusion criteria were the following: dysphagia, active cancer, severe cognitive impairment (assessed with a Mini-Mental State Examination score ≤ 9 points), -." A more specific description is recommended to allow the readers to understand the methods of selection of participants. For instance, were patients consecutively recruited? How was dysphagia defined and measured? Did all patients admit to the hospital routinely undergo dysphagia measurement and Mini-Mental State Examination? (If not, it is then quite likely that the recruited patient sample is a highly-selected subset of potentially eligible patient samples. The potential of biased observation could not be excluded. A discussion of this limitation in DISCUSSION is suggested.) In addition, please cite published references (if available) that describe the design, participants sampling and recruitment, and measurements of this study in more detail.
2. Materials and Methods: Length of stay and mortality are major variables in this study. I recommend describing in detail their definition and measurement in a separate section. In addition, regarding length of stay, it seems that the authors referred to time to hospital discharge. If yes, I suggest explicitly stating so. Regarding 30-day mortality, "number of deaths were recorded during hospitalization and at 30-day to hospital admission" (Line 69). How would the authors define the survival status of a patient who was discharged before Day 30? Would the authors classify that patient as not having 30-day mortality? If yes, how could the authors be certain that the patient would not die after being discharged? If the authors could not be certain, the 30-day mortality defined in this study would be misleading and cause much confusion. Therefore, I suggest that the authors focus only on in-hospital mortality in this study.
3. Materials and Methods (4. Statistical Analysis): "The multiple linear regression analysis using a enter model was performed to study the effect of independent variables (--) on LOS as dependent variable." However, the major drawback of this analytic method is that it fails to take into account censoring due to, for instance, death. I suggest using survival analysis (for instance, Cox regression) to estimate the effect of independent variables on time to discharge and using Kaplan–Meier curves to display and compare (using, for instance, log-rank test) the time to discharge of different groups (for instance, the group comparisons in Table 4 and Figure 1)
4. Results (Figure 1): The title read "Percentage of In-hospital mortality in frail-." However, the Y-Axis seems to be the length of stay. In addition, what does the "*" mean? Does it mean a statistically significant difference? An explanation in the figure legend is recommended.
5. Results (Figure 2): The title read "Length of stay (days) in all groups-." However, the Y-axis seems to be the percentage of mortality.
6. Discussion: A major limitation of this study is that the study population (246 patients) was a sample drawn from inpatients of a hospital. The results are susceptible to bias and may not be generalizable to other populations. A discussion of this limitation is suggested.
7. Discussion: Although the authors observed an association between CONUT score and length of stay, residual confounding (by, for instance, diseases and their severity) is still possible. A discussion of this limitation is suggested.
8. Discussion (Line 307): "-with a cross-sectional design, -." However, this study is prospective (longitudinal) in design.
9. Abstract (Line 10-11): "This study aims to investigate the predictive role of the CONUT score on clinical outcomes in frail patients-." Please specify the clinical outcome (for instance, length of stay and in-hospital mortality).
Minor English editing is needed.
Author Response
Comments:
- Materials and Methods (1.Study Population): " - as a prospective and observational single center study, based on a population aged 65 years or older recruited from –."The exclusion criteria were the following: dysphagia, active cancer, severe cognitive impairment (assessed with a Mini-Mental State Examination score ≤ 9 points), -." A more specific description is recommended to allow the readers to understand the methods of selection of participants. For instance, were patients consecutively recruited? How was dysphagia defined and measured? Did all patients admit to the hospital routinely undergo dysphagia measurement and Mini-Mental State Examination? (If not, it is then quite likely that the recruited patient sample is a highly-selected subset of potentially eligible patient samples. The potential of biased observation could not be excluded. A discussion of this limitation in DISCUSSION is suggested.) In addition, please cite published references (if available) that describe the design, participants sampling and recruitment, and measurements of this study in more detail.
Reply: we thank the reviewer for this valuable comment. Patients were consecutively recruited; this is now reported in the manuscript (line 51). MMSE is routinely administered to patients on admission to our hospital department. On the contrary, dysphagia is normally assessed on request by a trained logopaedics. We reported these statements in the methods section (lines 55-57). There are no published references on our study design, sampling and recruitment.
- Materials and Methods: Length of stay and mortality are major variables in this study.I recommend describing in detail their definition and measurement in a separate section. In addition, regarding length of stay, it seems that the authors referred to time to hospital discharge. If yes, I suggest explicitly stating so. Regarding 30-day mortality, "number of deaths were recorded during hospitalization and at 30-day to hospital admission" (Line 69). How would the authors define the survival status of a patient who was discharged before Day 30? Would the authors classify that patient as not having 30-day mortality? If yes, how could the authors be certain that the patient would not die after being discharged? If the authors could not be certain, the 30-day mortality defined in this study would be misleading and cause much confusion. Therefore, I suggest that the authors focus only on in-hospital mortality in this study.
Reply: we are grateful to the reviewer for this suggestion. We described LOS and mortality in a separate section of the methods. We defined the survival status of patients discharged before day 30 by telephone call. This was also described (lines 98-102).
- Materials and Methods (4. Statistical Analysis):"The multiple linear regression analysis using a enter model was performed to study the effect of independent variables (--) on LOS as dependent variable." However, the major drawback of this analytic method is that it fails to take into account censoring due to, for instance, death. I suggest using survival analysis (for instance, Cox regression) to estimate the effect of independent variables on time to discharge and using Kaplan–Meier curves to display and compare (using, for instance, log-rank test) the time to discharge of different groups (for instance, the group comparisons in Table 4 and Figure 1)
Reply: we appreciated the reviewer’ suggestion. Accordingly, we performed survival analysis using a Cox proportional hazard model, and generated Kaplan-Meier curves to represent time to discharge in all groups. The main text was consequently modified (lines 119-120; lines 247-262; table 9; figure 4).
- Results (Figure 1):The title read "Percentage of In-hospital mortality in frail-." However, the Y-Axis seems to be the length of stay. In addition, what does the "*" mean? Does it mean a statistically significant difference? An explanation in the figure legend is recommended.
Reply: We thank the reviewer for his observation. Indeed, Figure 1 and Figure 2 were wrongly reversed in the order of their inclusion in the text. We put the figures in the exact order.
- Results (Figure 2):The title read "Length of stay (days) in all groups-." However, the Y-axis seems to be the percentage of mortality.
Reply: Please consider the reply to the previous comment.
- Discussion:A major limitation of this study is that the study population (246 patients) was a sample drawn from inpatients of a hospital. The results are susceptible to bias and may not be generalizable to other populations. A discussion of this limitation is suggested.
Reply: we discussed this limitation, as suggested (lines 319-321).
- Discussion:Although the authors observed an association between CONUT score and length of stay, residual confounding (by, for instance, diseases and their severity) is still possible. A discussion of this limitation is suggested.
Reply: we discussed this limitation, as suggested (lines 317-319).
- Discussion (Line 307): "-with a cross-sectional design, -." However, this study is prospective (longitudinal) in design.
Reply: we modified the sentence as suggested (lines 313-314).
- Abstract (Line 10-11): "This study aims to investigate the predictive role of the CONUT score on clinical outcomes in frail patients-." Please specify the clinical outcome (for instance, length of stay and in-hospital mortality).
Reply: abstract was modified as suggested (line 11).
Round 2
Reviewer 1 Report
The authors have taken all comments into account
Reviewer 2 Report
All the comments have been appropriately addressed in this version of the manuscript.